# Gut Microbiota, Human Blood Metabolites, and Esophageal Cancer: A Mendelian Randomization Study

**DOI:** 10.3390/genes15060729

**Published:** 2024-06-02

**Authors:** Xiuzhi Li, Bingchen Xu, Han Yang, Zhihua Zhu

**Affiliations:** 1State Key Laboratory of Oncology in South China, Department of Thoracic Surgery, Sun Yat-sen University Cancer Center, Guangdong Provincial Clinical Research Center for Cancer, Collaborative Innovation Center for Cancer Medicine, Guangzhou 510060, China; lixz1@sysucc.org.cn; 2State Key Laboratory of Oncology in South China, Department of Minimally Invasive Intervention, Sun Yat-sen University Cancer Center, Guangdong Provincial Clinical Research Center for Cancer, Collaborative Innovation Center for Cancer Medicine, Guangzhou 510060, China; xubc@sysucc.org.cn; 3State Key Laboratory of Oncology in South China, Department of Thoracic Oncology, Sun Yat-sen University Cancer Center, Guangdong Provincial Clinical Research Center for Cancer, Collaborative Innovation Center for Cancer Medicine, Guangzhou 510060, China

**Keywords:** esophageal cancer, gut microbiota, human blood metabolites, Mendelian randomization analysis, mediation analysis

## Abstract

Background: Unbalances in the gut microbiota have been proposed as a possible cause of esophageal cancer (ESCA), yet the exact causal relationship remains unclear. Purpose: To investigate the potential causal relationship between the gut microbiota and ESCA with Mendelian randomization (MR) analysis. Methods: Genome-wide association studies (GWASs) of 207 gut microbial taxa (5 phyla, 10 classes, 13 orders, 26 families, 48 genera, and 105 species) and 205 gut microbiota metabolic pathways conducted by the Dutch Microbiome Project (DMP) and a FinnGen cohort GWAS of esophageal cancer specified the summary statistics. To investigate the possibility of a mediation effect between the gut microbiota and ESCA, mediation MR analyses were performed for 1091 blood metabolites and 309 metabolite ratios. Results: MR analysis indicated that the relative abundance of 10 gut microbial taxa was associated with ESCA but all the 12 gut microbiota metabolic pathways with ESCA indicated no statistically significant association existing. Two blood metabolites and a metabolite ratio were discovered to be mediating factors in the pathway from gut microbiota to ESCA. Conclusion: This research indicated the potential mediating effects of blood metabolites and offered genetic evidence in favor of a causal correlation between gut microbiota and ESCA.

## 1. Introduction

A major global disease, esophageal cancer (ESCA), ranks sixth among all cancers in terms of mortality based on global cancer statistics [1]. Individuals with early-stage ESCA may not recognize the symptoms of obstruction or stricture due to the dilated and muscular nature of the esophagus. Symptoms only appear after the tumor has progressed locally or even metastatically [2]. The majority of patients with ESCA in the United States and Europe are diagnosed with locally advanced or metastatic disease, which is ineligible for curative treatment [3]. Over 70% of patients in the UK receive a diagnosis of distant metastases or lymph node metastases, of which distant metastases account for about 40% [3]. Patients with advanced ESCA typically encounter an unfavorable prognosis. Due to etiological, molecular, and histological heterogeneity, advanced ESCA patients often acquire innate resistance to systemic therapy, significantly reducing treatment efficacy [4]. Surgical intervention remains the most effective way to treat ESAC, but even after surgery, less than 25% of individuals with advanced ESCA survive for five years [5]. Early adoption of preventative strategies and a detailed understanding of the etiology of ESCA is critical in reducing the incidence of ESCA.

With approximately 10^14^ species of microorganisms, the intestinal microbiota is regarded as the largest microbial reservoir in our body [6]. As an important regulator of human health, the gut microbiota plays an integral role in the development of the human immune system and the maintenance of intestinal homeostasis [7]. An increasing corpus of studies has demonstrated the intricate relationships between ESCA and the human gut flora in recent years [8,9]. The composition and abundance of fecal microorganisms in ESCA patients are closely related to the severity of the disease [10]. In addition, the genome-wide methylation level of ESCA can be regulated by the gut microbiota, which affects the occurrence, development, and metastasis of ESCA. The intestinal microbiota can function through various bioactive metabolites that systematically affect the internal microenvironment, including bile acids, short-chain fatty acids, and lipopolysaccharides, to regulate the function of the corresponding target organs [11,12]. Deficiency or disorder of intestinal flora significantly affects polysaccharide decomposition and lipid absorption, resulting in liver and adipose tissue dysfunction, leading to cardiovascular and cerebrovascular diseases, type 2 diabetes, and obesity, among other metabolic-related diseases [13]. Recent studies have shown that there were notable variations in the concentration of amino acids such as tryptophan and tyrosine, as well as lipids such as oleic acid and palmitoleic acid, between ESCA patients and healthy controls [14,15]. The evidence suggests that human circulating metabolites are integral components in the development of ESCA. Tumor cells can disrupt the entire metabolism in the process of continuously adapting to the dynamic metabolic microenvironment, thus affecting the distribution and content of metabolites in blood circulation, such as upregulating the glycolytic pathway under adequate oxygen conditions, resulting in rapid growth [16].

In recent years, for determining possible causal associations between various exposures and clinical outcomes, Mendelian randomization (MR) analysis has become widely applied, which is a method for determining the causal relationship between genetically predicted exposure and genetically predicted outcome, particularly single nucleotide polymorphisms (SNPs) [17]. Most epidemiologic investigations that have been accomplished on the causality between ESCA, human blood metabolites, and gut microbiota are based on conventional approaches (e.g., cross-sectional, case-control, cohort). Nevertheless, the estimates of effect may be impacted by a number of constraints, including confounding and reverse causation bias [18]. In recent years, for determining possible causal associations between various exposures and clinical outcomes, MR analysis has become widely applied. By virtue of the fact that allelic randomization occurs prior to the onset of disease, MR analysis has an advantage over conventional observational studies in reducing reverse causation bias. Furthermore, the independent assortment and random segregation of genetic polymorphisms at conception mitigate the confounding bias, as genetic markers are used as instrumental variables (IVs) in MR analysis [19].

Given the absence of research examining the causal correlation between gut microbiota and ESCA mediated by human blood metabolites, we undertook a two-sample, two-step MR analysis to reveal the relationship.

## 2. Method

### 2.1. Study Design

Figure 1 illustrates the study design. More than 647,920 participants were selected from summary level, publicly available datasets to conduct a large two-step, two-sample MR study using a two-step strategy to assess the association between the genetic prediction of gut microbiota and esophageal carcinoma and to determine whether plasma metabolites could mediate this association. A two-sample MR analysis utilizing different datasets is performed to assess correlations of the same genetic variants with exposure (e.g., gut microbiota) and outcome (e.g., esophageal carcinoma). Initially, the causal effects of genetic prediction of 412 gut microbiota with a genetic disposition to esophageal carcinoma were analyzed, and the two-step approach utilized in mediating analysis examined the association between genetically predicted gut microbiota and each potential mediator. Subsequently, we investigate and quantify the mediation effects of potential mediators in the pathway from the 412 gut microbiota to esophageal carcinoma.

### 2.2. Data Sources

#### 2.2.1. Genetic Instrumental Variables for Gut Microbiome

A large-scale genome-wide association study (GWAS) carried out by the Dutch Microbiome Project (DMP) provided the species-level dataset for the gut microbiota [20]. 7738 participants of European descent were involved in the analysis of this dataset, which is the hitherto largest species-level genomics study on the human gut microbiota. An analysis of stool samples was performed utilizing shotgun metagenomic sequencing to determine the gut microbiome, ultimately identifying 207 taxonomies (105 species, 48 genera, 26 families, 13 orders, 10 classes, 5 phyla) and 205 gut microbiota metabolic pathways related to microbial functions. This GWAS dataset is described in more detail in its original publication [20]. The GWAS data are publicly available at https://mibiogen.gcc.rug.nl, accessed on 5 January 2024.

SNPs with genome-wide significance (*p* < 1 × 10^−5^) and clumping at a linkage disequilibrium (LD) threshold of r^2^ < 0.001 (clumping distance: 10,000 kb) were used in the analyses as instrumental variables (IVs) for gut microbiota. The estimated F-statistics for exposure, which were used to quantify the IVs, ranged from 19 to 57, which is consistent with the notion of F > 10 for MR studies and enables us to remove weak instrumental variable bias [21].

#### 2.2.2. Genetic Instrumental Variables for Potential Mediators

A recent GWAS carried out on the Canadian Longitudinal Study on Aging (CLSA) cohort, involving a total of 8299 individuals and approximately 15.4 million SNPs, explored the association of SNPs with human metabolite levels. A genome-wide association study of 1091 blood metabolites and 309 metabolite ratios was performed in this research [22]. The results of the present investigation identified correlations with 248 loci containing 690 metabolites and 69 loci containing 143 metabolite ratios. Kyoto Encyclopedia of Genes and Genomes database is the basis for the classification of these known metabolites into categories such as peptide, nucleotide, amino acid, carbohydrates, cofactors, and vitamin, energy, lipid, and xenobiotics metabolism.

#### 2.2.3. Genetic Instrumental Variables for Esophageal Carcinoma

Esophageal carcinoma GWAS summary data were derived from the tenth version of the FinnGen consortium (https://r10.finngen.fi/, accessed on 5 January 2024). Esophageal carcinoma was identified using International Classification of Diseases (ICD) diagnosis codes in this prospective cohort study, involving 619 cases and 314,193 controls originating from European ancestry. FinnGen has correlated genetic variation with data on healthy individuals to uncover disease mechanisms and genetic predispositions [23].

### 2.3. Statistical Analyses

We examined the potential association between genetically predicted gut microbiota and genetically predicted esophageal carcinoma by applying a bidirectional two-sample MR analysis. Additionally, to examine the possible mediation effects of human blood metabolites in the causal relationship, a two-step MR analysis was carried out using summary statistical data.

In both forward and reverse directions of MR analyses, the inverse-variance weighted (IVW) method was utilized as the primary analytical approach to estimate odds ratios and *p*-values, widely recognized as the most robust methodology for generating reliable causal estimates in MR studies.

The IVW method, analyzing the causal effects of exposure SNPs on outcome data, was employed as the primary approach [24]. *p*-values of IVW less than 0.05 and consistent directions for both IVW and MR–Egger demonstrated statistical significance in the results. A two-sided *p*-value that was accepted after the Bonferroni adjustment *p*-values of 0.0001 (0.05/412) for gut microbiota and 0.00004 (0.05/1400) for metabolites were deemed statistically significant, while *p* < 0.05 was regarded as indicating a suggestively significant association. In the absence of effective instruments, the weighted median method was employed, as it is capable of offering reliable causal effect estimates even if less than fifty percent of the information is derived from valid instruments [25]. To discover the anomalies in the analysis due to the large horizontal pleiotropy during the MR analysis and to account for the weak effects and uncertainties of the weak horizontal pleiotropy, we performed further analysis using Bayesian weighted Mendelian randomization (BWMR) [26].

### 2.4. Sensitivity Analyses

An assessment of the heterogeneity between SNPs was carried out using Cochran’s Q statistics [27]. Unless evidence of substantial heterogeneity (*p* < 0.05), fixed-effects models were employed; otherwise, random-effects models were applied. As well as determining whether instrumental SNPs are multi-effect, we used the MR–Egger method to identify the multi-effects. The *p* value of its intercept was calculated as part of an MR–Egger regression analysis for uncovering possible horizontal pleiotropy [28]. Furthermore, we also performed MR pleiotropy residual sum and outlier (MR-PRESSO), thus removing possible outliers from multi-effects estimates [29] and rectifying potential confounding factors [30]. The odds ratio (OR) and 95% confidence interval (CI) per standard deviation were calculated as the result. The mediation proportions were determined based on the formula: (beta1 × beta2) / beta_all. beta_all represents the total causal effects of gut microbiota on esophageal carcinoma derived from the main analysis, beta1 represents the estimated effect of gut microbiota-related traits on potential blood metabolites mediators, and beta2 represents the causal effects of blood metabolites mediators on esophageal carcinoma.

## 3. Results

### 3.1. Bidirectional Two-Sample MR Analyses between Gut Microbiota and Esophageal Carcinoma

In total, ten gut microbiota (including one phylum, one family, two genera, and six species) and twelve gut microbiota metabolic pathways were associated with esophageal carcinoma. In Appendix A, 85 SNPs for 10 gut microbiota and 117 SNPs for 12 gut microbiota metabolic pathways are presented in detail.

Based on the MR analyses, Figure 2 illustrates the correlation of four gut microbiota with the increased risk of esophageal carcinoma. The genus *Phascolarctobacterium* (OR = 1.426, 95%CI = 1.092 ~ 1.862, *p* = 0.009), species *Phascolarctobacterium succinatutens* (OR = 1.426, 95%CI = 1.093 ~ 1.861, *p* = 0.009), species *Bifidobacterium adolescentis* (OR = 1.426, 95%CI = 1.012 ~ 2.139, *p* = 0.043) and phylum *Proteobacteria* (OR =  1.724, 95%CI = 1.132 ~ 2.626, *p* = 0.011) significantly increased the risk of esophageal carcinoma. Six genetically predicted gut microbiotas were associated with the decreased risk of esophageal carcinoma. The family *Ruminococcaceae* (OR = 0.446, 95%CI = 0.258 ~ 0.770, *p* = 0.004), species *Streptococcus thermophilus* (OR = 0.586, 95%CI = 0.402 ~ 0.855, *p* = 0.006), species *Clostridium leptum* (OR = 0.621, 95%CI = 0.436 ~ 0.885, *p* = 0.008), genus *Erysipelotrichaceae* no name (OR = 0.716, 95%CI = 0.552 ~ 0.930, *p* = 0.012), species *Eubacterium hallii* (OR = 0.719, 95%CI = 0.532 ~ 0.973, *p* = 0.033), and species *Holdemania* unclassified (OR = 0.687, 95%CI = 0.504 ~ 0.937, *p* = 0.018) remarkably decreased the risk of esophageal carcinoma. The BWMR method yielded consistent conclusions in the causal association analysis of these ten gut microbiota. However, the results of the weighted median method with genus *Erysipelotrichaceae* no name (*p* = 0.073), species *B. adolescentis* (*p* = 0.134), species Eubacterium hallii (*p* = 0.323), and species *Holdemania* unclassified (*p* = 0.119) were negative, and no mediating metabolite was found to be associated with species *S. thermophilus* and phylum Proteobacteria, leading to their exclusion. The reverse MR analysis revealed no significant causal effects of genetic prediction of esophageal carcinoma on the 4 gut microbiota as mentioned above with the *p* value higher than 0.05 shown in the IVW method, indicating the absence of a reverse causal relationship between them (Appendix A). Ultimately, the family *Ruminococcaceae*, genus *Phascolarctobacterium*, species *C. leptum,* and species *P. succinatutens* were chosen as the exposure variables (Appendix A).

Only the causal association between the PANTOSYN.PWY..pantothenate.and. coenzyme.A.biosynthesis.I pathway and ESCA was validated by both the IVW approach and the weighted median method among the 12 gut microbiota metabolic pathways; however, these pathways were excluded due to the absence of any significant mediated metabolites (Appendix A).

### 3.2. Causal Effects of the Selected Gut Microbiota on the Human Blood Metabolites

Figure 3 identifies that family *Ruminococcaceae* was causally associated with higher 1-arachidonoyl-gpc (20:4n6) levels (β = 0.265, 95%CI = 0.103 ~ 0.427, *p* = 0.001), phosphate levels (β = 0.302, 95%CI = 0.143 ~ 0.461, *p* = 0.0002), X-23648 levels (β = 0.274, 95%CI = 0.105 ~ 0.443, *p* = 0.001), phosphate-to-glucose ratio (β = 0.278, 95%CI = 0.120 ~ 0.437, *p* = 0.0006), and Arachidonate (20:4n6)-to-caffeine ratio (β = 0.262, 95%CI = 0.091 ~ 0.433, *p* = 0.003).

We identified that the family *Ruminococcaceae*, genus *Phascolarctobacterium*, species *C. leptum*, and species *P. succinatutens* were causally associated with 69, 59, 47, and 61 metabolites, respectively, primarily applying the IVW approach (Appendix A). The weighted median method and BWMR supported the robustness of the results.

### 3.3. Bidirectional Two-Sample MR Analyses between Human Blood Metabolites and Esophageal Carcinoma

After examining the causal association between gut microbiota and the above significant metabolites through the weighted median method and BWMR, we found that perfluorooctanoate (PFOA) levels (OR = 0.713, 95%CI = 0.508 ~ 1.000, *p* = 0.0498) were a significant risk factor in the causal pathway from species *C. leptum* to esophageal carcinoma; the cholate-to-bilirubin (Z, Z) ratio (OR = 1.298, 95%CI = 1.010 ~ 1.668, *p* = 0.041) was a significant risk factor in the causal pathway from genus *Phascolarctobacterium* to esophageal carcinoma; the cholate-to-bilirubin (Z, Z) ratio (OR = 1.298, 95%CI = 1.010 ~ 1.668, *p* = 0.041) was also a significant risk factor in the causal pathway from species *P. succinatutens* to esophageal carcinoma. Genetic prediction of 1-arachidonoyl-gpc (20:4n6) levels (OR = 0.814, 95%CI = 0.664 ~ 0.997, *p* = 0.047) was a significant risk factor in the causal pathway from family *Ruminococcaceae* to esophageal carcinoma as shown in Figure 4. However, the causal relationship between family *Ruminococcaceae* and 1-arachidonoyl-gpc (20:4n6) levels was not identified by the weighted median method with a *p* value higher than 0.05. MR analysis of the rest of the four metabolites with esophageal carcinoma indicated no statistically significant association existing. The detailed information on the results is presented in Appendix A. Reverse MR analysis indicated no reverse causal association between the 3 blood metabolites and esophageal carcinoma (Appendix A).

### 3.4. Mediation Effects of the Selected Human Blood Metabolites on Esophageal Carcinoma

For the mediation analysis illustrated in Figure 5, we excluded mediating factors that were not causally affected by gut microbiota and those that did not causally influence esophageal carcinoma. Finally, our results indicated that PFOA levels, cholate-to-bilirubin (Z, Z) ratio, and 1-arachidonoyl-gpc (20:4n6) levels were significant risk factors mediating the correlation of gut microbiota-related traits with esophageal carcinoma. The overall effect can be separated into direct effect (via mediators) and indirect effect (without mediators). Our results demonstrated that PFOA levels accounted for 9.74% in the causal pathway from species *C. leptum* to esophageal carcinoma; the cholate-to-bilirubin (Z, Z) ratio accounted for 11.45% in the causal pathway from genus Phascolarctobacterium to esophageal carcinoma; the cholate-to-bilirubin (Z, Z) ratio accounted for 11.42% in the causal pathway from species *P. succinatutens* to esophageal carcinoma; and 1-arachidonoyl-gpc (20:4n6) levels accounted for 6.75% in the causal pathway from family *Ruminococcaceae* to esophageal carcinoma.

### 3.5. Sensitivity Analyses

There is a low likelihood of weak instrument bias for these instrumental factors, as shown by the F-statistics for the selected SNPs, which are all over 10 (Appendix A). There is no LD and the SNPs are randomly dispersed, according to the r^2^ values, which range from zero to one (Appendix A). To assess the heterogeneity of our estimates, we calculated Cochrane’s Q and *p* values derived from Cochrane’s Q test (Appendix A). No evidence of significant heterogeneity was found in our analysis. To test and correct for the directional pleiotropy in causal estimates, a series of sensitivity analyses were carried out. The null results of the directional pleiotropy were indicated by the other MR analyses stated above, which did not find any significant intercept. Furthermore, a leave-one-out analysis was carried out to ascertain whether a particular SNP substantially deviated from the causal estimate, evaluating the effect of each SNP on the overall causal estimate (Appendix A). All of our positive results were consistent after removing the outliers in the original MR-PRESSO global test, which was utilized to ascertain and exclude outliers, as well as decrease heterogeneity in our analysis (Appendix A).

## 4. Discussion

To the best of our knowledge, based on statistical approaches that account for directional pleiotropy, this is the first study to investigate the likelihood of metabolite traits mediating a causal path between gut microbiota and ESCA. In this comprehensive and large-scale MR analysis, we affirmed that PFOA levels and 1-arachidonoyl-gpc (20:4n6) levels, respectively, mediate the causal influence of species *C. leptum* and family *Ruminococcaceae* on ESCA, while the cholate-to-bilirubin (Z, Z) ratio mediates the pathway from genus *Phascolarctobacterium* and species *P. succinatutens* to ESCA.

There is a large microbial population in the stool of adults, of which Cluster IV (*C. leptum* group) occupies a dominant position, and its abundance is generally higher than 15%. These bacteria tend to be associated with multiple metabolic pathways in the body. These bacteria are involved in a variety of metabolic pathways that maintain the balance of the intestinal microecological environment. One of the main sources of energy for colonic epithelial cells to regulate intestinal epithelial function is short-chain fatty acids (SCFAs) produced by *C. leptum* [31,32]. *C. leptum* ferments polysaccharides through acetyl-CoA and pyruvate pathways to produce propionate and butyrate [33], thereby controlling glucose concentration in the intestinal microenvironment. Recent studies have shown that the intestinal flora produces some metabolites with weight loss effects in the process of fermenting polysaccharides [34]. Li et al. found that *C. leptum* can alleviate obesity by fermenting metabolites produced by FP [35], which could lower the risk factors of ESCA [36]. It is plausible to speculate that this may be one of the mechanisms by which *C. leptum* can lower the risk of ESCA.

PFOA, one of the four types of polyfluoroalkyl substances (PFASs), is a newly discovered environmental contaminant that can cause health problems as an endocrine disruptor [37]. Four PFAS (PFOA, perfluorooctane sulfonic acid [PFOS], perfluorohexane sulfonic acid [PFHxS], and perfluorononanoic acid [PFNA]) have been reported to be detected in the serum among individuals over 12 in the United States, with a positive rate of more than 98%, indicating the prevalence of PFOA exposure [38]. Several studies have elucidated the mechanism of activation of PFOA in the development of ESCA [39].

High structural similarity exists between SCFAs and PFAS, which have been demonstrated to interfere with hepatic lipid metabolism through interactions with a variety of nuclear receptors, including peroxisome proliferator-activated receptors (PPARs) [40]. Additionally, glucose metabolism pathways may be affected [41]. High levels of PFOA exposure have been correlated to an unbalance in Clostridium abundance, according to recent studies [42]. These findings imply that PFOA may be employed as a potential mediator to affect Clostridium’s causal effect on ESCA.

Phascolarctobacterium is a fecal-phase intestinal bacterium extracted from koala excrement by Del Dot et al. [43]. This bacterium is classified as Gram-negative, pleomorphic rod-shaped cells composed of *P. faecium* and *P. succinatutens* [44]. A recent study has shown that Phascolarctobacterium is widespread in the human gastrointestinal tract and can produce SCFAs, including acetic acid, propionic acid, isobutyric acid, butyric acid, and isovaleric acid [45]. Phascolarctobacterium stimulates growth by succinic acid and decomposes it into propionic acid [46], which is involved in significant metabolic pathways, such as hepatic gluconeogenesis [47]. Clinically substantial reductions in Phascolarctobacterium abundance are observed in patients with head and neck cancer [48], while pancreatic and prostate cancer patients exhibit significantly larger abundances compared to healthy controls [49,50].

A bile acid receptor, the G-protein coupled bile acid receptor Gpbar1 (TGR5), is widely distributed in muscles, adipose tissue, immune systems, enteric nervous systems, etc., which can modulate the expression of TGR5 in the EAC FLO cell line and the BE BAR-T cell line [51], and possibly affect the progression from BE to EAC [52,53]. One of the bile acid receptors known as the vitamin D receptor (VDR) is overexpressed in precancerous lesions and EAC [54]. These findings imply that bile acids may be involved in the early carcinogenesis process through TGR5 and VDR. Bilirubin levels in serum reveal liver dysfunction as a result of chronic viral hepatitis, alcohol consumption, and chemoradiotherapy. The albumin–bilirubin (ALBI) score was proved to be a predictive prognostic factor in patients with EACC, as the 5 years survival rate in the albumin–bilirubin ratio low group was significantly higher than that in the albumin-bilirubin ratio high group [55,56].

New insights into the relationship between the bile acid and gut microbiota have highlighted the interplay between intestinal bacteria and bile acids in controlling digestive health [57]. A slight variation in bile acids can cause a significant shift in the composition of the bacterial community, which advantageously aids in the host’s defense against infections [58]. Phascolarctobacterium abundance was discovered to vary in post-cholecystectomy diarrhea (PCD) patients, and Xu et al. verified that there is a positive correlation between Phascolarctobacterium and taurolithocholic acid (TLCA) [59].

In healthy individuals, the colonic mucosal biofilm contains Ruminococcaceae bacteria which are strictly anaerobic [60]. SCFAs generated by Ruminococcaceae metabolism can stabilize the homeostasis of the intestinal microenvironment [45]. Dysfunction of the colonic mucosa frequently coexists with osmotic diarrhea and is generally brought on by a deficiency in SCFAs [61]. Butyrate exerts anti-inflammatory effects by upregulating the tight junctions between colonocytes to strengthen the intestinal barrier and prevent lipopolysaccharides (LPSs) from being transported into the systemic circulation [62]. Therefore, the abundance of Ruminococcaceae typically declines in patients with inflammatory bowel diseases such as Crohn’s disease or ulcerative colitis [32,63,64]. As an essential lysophosphatidylcholine, 1-Arachidonoyl-GPC hinders the migration of CXCR3+ T cells to the inflammatory microenvironment by inhibiting autoimmunity [65,66]. According to the Phe-MR analysis of 655 diseases conducted by Jia et al., 1-arachidonol-gpc supplementation may lead to an increased risk of benign neoplasm of the colon and impaired thyroid function [67]. The mechanism of 1-arachidonol-GPC in the prevention of ESCA is still poorly understood, and research on it should be further conducted.

Overall, despite some evidence supporting the association of species *C. leptum*, genus Phascolarctobacterium, species *P. succinatutens,* and family Ruminococcaceae with ESCA, the evidence is apparently inadequate and of relatively low quality. Hence, clinical trials with larger sample sizes, as well as studies of cellular mechanisms are necessary to confirm the health effects and mechanisms underlying these bacteria.

Several crucial strengths have been identified in our study. First, our study closes a knowledge vacuum in this area by examining whether gut microbiota are causally associated with ESCA through metabolite traits, as no other study has done so yet. Second, to move forward with animal experiments and mechanism research, we analyzed gut microbiota at the species level. Finally, as part of our attempt to uncover the possible mechanisms responsible for the association between gut microbiota and ESCA, we employed a mediating analysis.

However, we must acknowledge certain limitations in our research. First, since our analysis was carried out mostly among European populations, the results should be extrapolated with caution since the correlation between the gut microbiota and the host genomes may vary based on ethnicity. Second, as the gut microbiota of different populations vary considerably in terms of their gut microbiota composition, the sample size of the GWAS summary data may not have been adequate for all potential causal relationships to be revealed. Third, our study identified independent variants associated with gut microbiota traits at the genome-wide significance of *p* < 1 × 10^−5^, as the criteria applied in the primary GWAS and MR analysis of gut microbiota was found in the other literature. However, as indicated by large F-statistics, genetic instruments were significantly correlated to exposure in this study.

## 5. Conclusions

We found four gut microbiota in our MR analysis that may be causally related to ESCA. Our research offers genetic evidence that alterations in the gut flora could be an essential risk factor for the progression of ESCA, which could be mediated by several human blood metabolites. These findings provide novel perspectives on the pathogenesis of ESCA and propose possible EACA intervention targets. To validate these results and comprehend the underlying mechanisms involved, further investigation is required.

## Figures and Tables

**Figure 1 genes-15-00729-f001:**
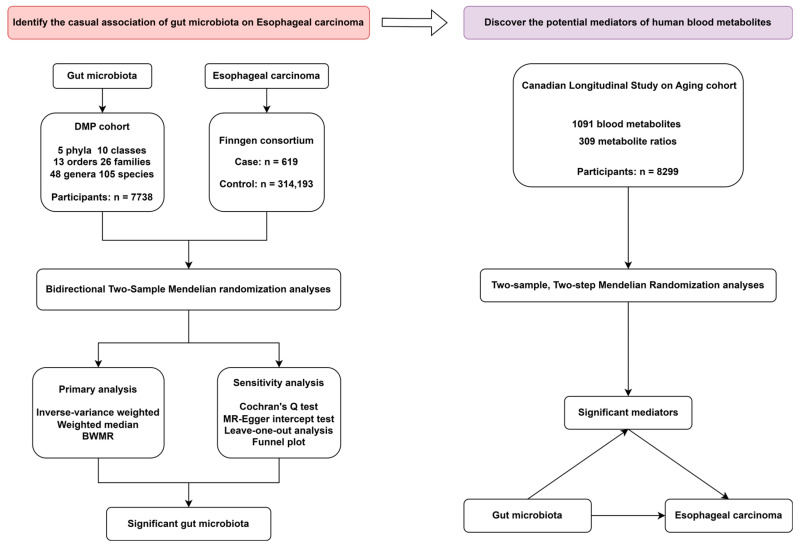
Study Design. The overview of our two-stage study design is displayed in the diagram. First, to discover putative causal gut microbiota of esophageal cancer, we employed a bidirectional two-sample Mendelian randomization (MR) study, along with several sensitivity analyses. Second, an MR analysis of mediation was carried out. We assessed the causal association of several human blood metabolites on gut microbiota, as well as the degree to which these blood metabolites modulate the influence of gut microbiota on esophageal cancer.

**Figure 2 genes-15-00729-f002:**
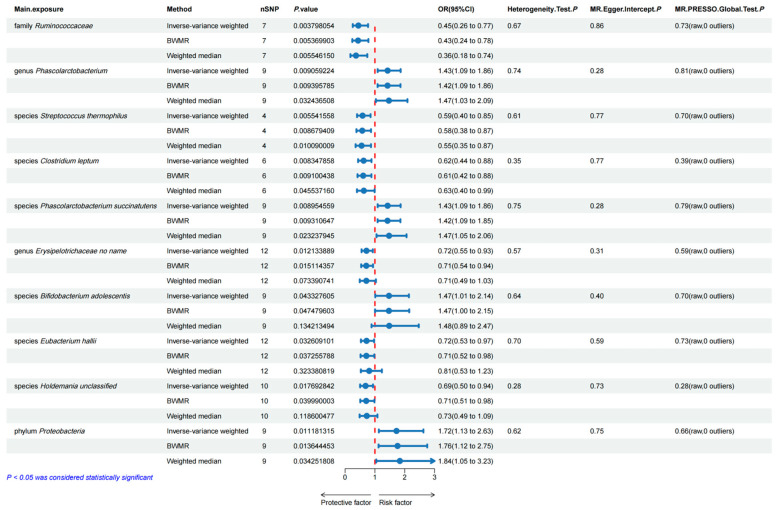
Mendelian randomization analysis of causal effects between vital gut microbiota and esophageal cancer.

**Figure 3 genes-15-00729-f003:**
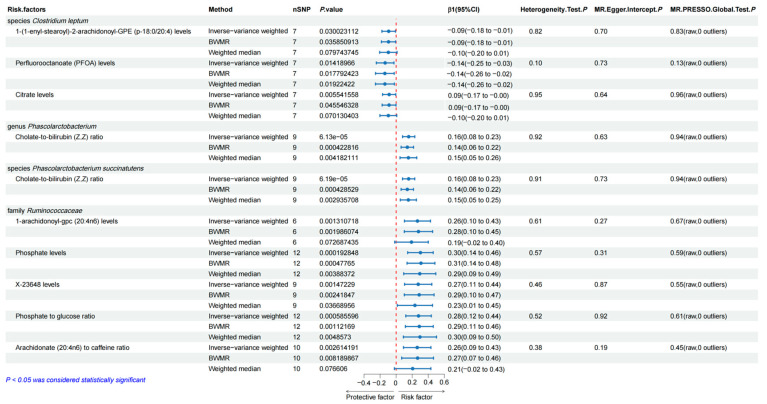
Mendelian randomization analysis of causal effects between vital gut microbiota and mediated blood metabolites.

**Figure 4 genes-15-00729-f004:**
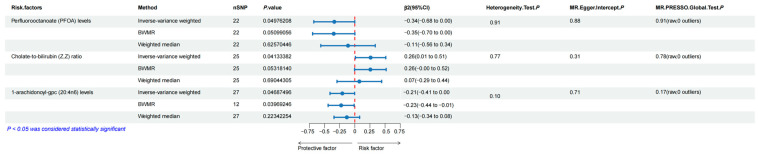
Mendelian randomization analysis of causal effects between mediated blood metabolites and esophageal cancer.

**Figure 5 genes-15-00729-f005:**
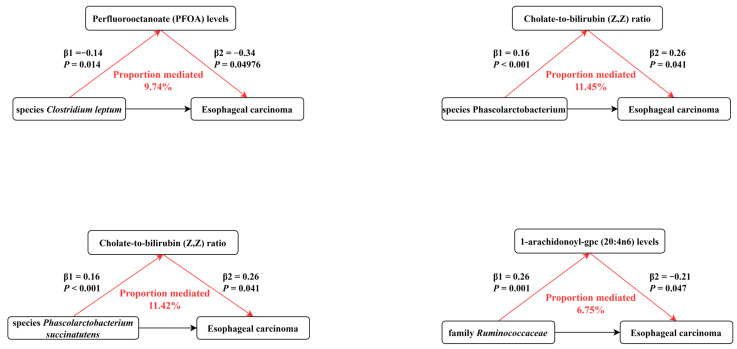
The proportions of each significant blood metabolite mediating from corresponding gut microbiota to esophageal cancer.

## Data Availability

GWAS summary data for gut microbiota from the Dutch Microbiome project are available at https://dutchmicrobiomeproject.molgeniscloud.org/, accessed on 5 January 2024; the GWAS summary statistics of human blood metabolites was obtained from the GWAS Catalog (https://www.ebi.ac.uk/gwas/, accessed on 5 January 2024); GWAS summary statistics for esophageal cancer is available at (https://www.finngen.fi/en/, accessed on 5 January 2024) access results.

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
