# Peer review of "Gut Microbiota, Human Blood Metabolites, and Esophageal Cancer: A Mendelian Randomization Study"

_genes, 2024, doi:10.3390/genes15060729_

Round 1

Reviewer 1 Report

Comments and Suggestions for Authors

In this manuscript titled “Gut microbiota, human blood metabolites, and esophageal cancer: a Mendelian randomization study,” the authors aimed to use Mendelian randomization analysis to explore the potential causal link between gut microbiota and esophageal cancer. The research found associations between certain gut microbial taxa and esophageal cancer, as well as identified blood metabolites as potential mediators in the pathway from gut microbiota to esophageal cancer, providing genetic evidence for a possible causal relationship.

The key findings and analyses presented in the manuscript could benefit from enhancements.

Major

----

* The mediation analysis in this study may introduce some bias. The author designed the two-step MR analysis for mediation detection based on the hypothesis that the gut microbiota potentially caused ESCA through the mediations of metabolites. However, the interplay between gut microbiota, small molecules and host is typically complex. Mediation direction should matter during this process. How does the gut microbiota perform from metabolites to ESCA?

* Relatively, the discovery of new pathways from gut microbiota to ESCA may be limited in the current mediation analysis. The authors started with the selected potential “causal” microbial taxa for mediation, which may miss some pathways in the causal links mediated by metabolites. It’s possible that some taxa were not directly linked to ESCA, but they could have a causal effect on ESCA indirectly through the meditation of metabolites. 

* The methodology needs more improvement and clarity:
 - Were the p-values corrected by the multiple testing adjustment in the analysis? Correcting for random events that erroneously appear significant is crucial. The detected associations should be performed with multi-testing correction.
 - More detailed explanations are needed in the Methods section. Which tools and what specific criteria were employed in the analysis? How did the two-step MR analysis perform?

Moderate

----

* More clarity is needed on how the authors removed pleiotropy causes and whether these causes were included in their presented results.

* It would be more helpful to describe how the authors use the published data in Section 2.2.2. So far it’s only an introduction of this cohort.

* There is a controversial description for microbial pathways in the first and third paragraphs of Section 3.1.1.

* In Fig 4, it might lead some confusion to present the same metabolite with different MR results without the context of associated taxa.

* Why did the authors claim “a positive association between gut microbiota and ESCA” in section 3.5? It looks like bidirectional associations (including both positive and negative).

Minor

----

* Something is messy in Section 2.2.3. The heading and paragraph is mismatched.

* Some texts are overlapped in Fig. 4.

Comments on the Quality of English Language

Tweaking the language of the manuscript is helpful to improve readability:

* The authors should go through the text and make it more consistent with regards to the terminology: microbiota, taxa, taxonomy. Currently these terms are scattered in the texts.
* Part of the text feels not completely to the point and repetitive. E.g. the second paragraph of Section 3.1.1 looks like using repeated sentences to describe the same thing. 

Author Response

Based on your sincere suggestions, we have made the following changes to the manuscript:

  1. We added to the discussion section about the influence of the gut microbiota on the occurrence of ESCA through related metabolites.
  2. We performed a Bonferroni adjustment for multi-testing correction.
  3. We have shown our selection criteria in more detail in the Methods section and the Results section.
  4. We have improved the description of heterogeneity and pleiotropy in Section 3.5.
  5. We have added a description of the data source.
  6. We have made changes to the contradictory paragraphs in Section 3.1.1 and Section 3.5.
  7. We've made changes to the text overlap and displayed details in Figure 4.
  8. We have changed the title of 2.2.3.

With regard to your suggestion that we may ignore the indirect effects that some microbiota may have through metabolites, we take the following view:

       One of the prerequisites for a mediated Mendelian analysis is that a two-sample analysis between exposure and outcome is positive, which is the total effect, not the direct effect (the direct effect is the total effect minus the indirect effect). We also searched the literature on mediating Mendelian mediators, which sought out relevant mediators from positive exposures [1,2,3].

References

  1. Dai H, Hou T, Wang Q, Hou Y, Zhu Z, Zhu Y, Zhao Z, Li M, Lin H, Wang S, Zheng R, Xu Y, Lu J, Wang T, Ning G, Wang W, Zheng J, Bi Y, Xu M. Roles of gut microbiota in atrial fibrillation: insights from Mendelian randomization analysis and genetic data from over 430,000 cohort study participants. Cardiovasc Diabetol. 2023 Nov 8;22(1):306. doi: 10.1186/s12933-023-02045-6. PMID: 37940997; PMCID: PMC10633980.
  2. Ji D, Chen WZ, Zhang L, Zhang ZH, Chen LJ. Gut microbiota, circulating cytokines and dementia: a Mendelian randomization study. J Neuroinflammation. 2024 Jan 4;21(1):2. doi: 10.1186/s12974-023-02999-0. PMID: 38178103; PMCID: PMC10765696.
  3. Wang Q, Dai H, Hou T, Hou Y, Wang T, Lin H, Zhao Z, Li M, Zheng R, Wang S, Lu J, Xu Y, Liu R, Ning G, Wang W, Bi Y, Zheng J, Xu M. Dissecting Causal Relationships Between Gut Microbiota, Blood Metabolites, and Stroke: A Mendelian Randomization Study. J Stroke. 2023 Sep;25(3):350-360. doi: 10.5853/jos.2023.00381. Epub 2023 Sep 26. PMID: 37813672; PMCID: PMC10574297.

Reviewer 2 Report

Comments and Suggestions for Authors

Dear,

I read with great interest this paper, which shows a potential causal relationship between the gut microbiota and ESCA with Mendelian randomization (MR) analysis. In general, the manuscript is well-written and well-presented. I just suggest to the authors to make a final figure with the main findings and their correlations to get more readers into it.

Comments on the Quality of English Language

Need to make a minor grammar corrections.

Author Response

Thank you for your suggestion, we have made some additions to Figure 5 based on your suggestion, it is a great honor to have your affirmation!